# Effect of Short Term Intensive Lifestyle Intervention on Hepatic Steatosis Indexes in Adults with Obesity and/or Type 2 Diabetes

**DOI:** 10.3390/jcm8060851

**Published:** 2019-06-14

**Authors:** Elisa Reginato, Roberto Pippi, Cristina Aiello, Emilia Sbroma Tomaro, Claudia Ranucci, Livia Buratta, Vittorio Bini, Giulio Marchesini, Pierpaolo De Feo, Carmine Fanelli

**Affiliations:** 1Healthy Lifestyle Institute, C.U.R.I.A.Mo (Centro Universitario Ricerca Interdipartimentale Attività Motoria), University of Perugia, Via G. Bambagioni, 19 06126 Perugia, Italy; elisa.reginato@gmail.com (E.R.); robertopippi@gmail.com (R.P.); cristina.aiello@hotmail.com (C.A.); emilia.sbroma@virgilio.it (E.S.T.); claudiaranucci.diet@gmail.com (C.R.); liviaburatta@hotmail.it (L.B.); pierpaolodefeo@gmail.com (P.D.F.); 2Department of Medicine, University of Perugia, 06126 Perugia, Italy; vittorio.bini@unipg.it; 3Unit of Metabolic Diseases and Clinical Dietetics, Alma Mater University of Bologna, 40126 Bologna, Italy; giulio.marchesini@unibo.it

**Keywords:** Non-alcoholic fatty liver disease, obesity, type 2 diabetes mellitus, lifestyle, exercise

## Abstract

Background: Non-alcoholic fatty liver disease (NAFLD) has an estimated prevalence of 20–30% in the general population and even higher in individuals with metabolic risk factors. The aim of this study was to evaluate the effect of a lifestyle intervention program on surrogate markers of hepatic steatosis in obesity and/or type 2 diabetes patients, enrolled in the C.U.R.I.A.Mo. (Centro Universitario di Ricerca Interdipartimentale Attività Motoria) trial. Methods: 102 subjects (56 females and 46 males, aged between 23 and 78) with type 2 diabetes, obesity or a BMI of at least 25 kg/m^2^ with comorbidities, participated in the intensive phase of a multidisciplinary lifestyle intervention program at the Healthy Lifestyle Institute of the University of Perugia (C.U.R.I.A.Mo.). Six indices related to NAFLD (Visceral Adiposity Index, Fatty Liver index, Non-Alcoholic Fatty Liver Disease liver fat score and liver fat equation, hepatic steatosis index and TyG index) were calculated before and after a three-month multidisciplinary lifestyle intervention. Results: The intervention improved the anthropometric and clinical parameters in the total population, the obese and/or diabetics. Data showed a significant weight loss, a reduced waist circumference, triglycerides, and an improvement in Mediterranean diet adherence. Hepatic steatosis indices were significantly reduced in the total population and in different subgroups (males, females, obesity and diabetes).

## 1. Introduction

Non-alcoholic fatty liver disease (NAFLD) is a significant public health problem, with an estimated prevalence of 20–30% in the general population and even higher in individuals with metabolic risk factors (visceral obesity, type 2 diabetes, dyslipidemia, and/or insulin resistance) [1]. 

Patients with NAFLD show higher all-cause mortality and increased risk for liver-related death and cardiovascular disease [2,3]. Due to the pathophysiological link existing between visceral adiposity, insulin resistance, and hepatic steatosis, therapeutic interventions aimed at improving lifestyle are essential and are recommended at any stage of disease [4]. Dietary changes are recommended by Clinical Practice Guidelines shared by the three major European Associations most consistently involved in the care of NAFLD/NASH patients recommend in their shared Clinical Pratice Guidelines weight loss of 7%, physical activity, reduced sedentary behavior, and dietary changes, all of which should be maintained throughout life [4].

The understanding of the overall prevalence of NAFLD is largely dependent on the criteria used for assessment. Liver biopsy represents the gold standard for diagnosis and the assessment of NAFLD severity. Although it can differentiate simple steatosis, steatohepatitis, and fibrosis, liver biopsy has certain drawbacks, as it is invasive, not feasible in all NAFLD patients, costly, suffers from sampling errors, has intra- and inter-observer variability, and must be limited to patients with increased risks of advanced fibrosis [5]. The mass screening of asymptomatic individuals using imaging modalities, such as ultrasonography and computed tomography, is not cost-effective because these studies are expensive. Hepatic ultrasound is highly operator-dependent and has limited repeatability and reproducibility [6] owing to subjective qualitative features of liver echogenicity. More precise imaging methodologies have been developed, including computed tomography (CT), magnetic resonance imaging (MRI), and spectroscopy (MRS); however, these are not widely available. CT provides objective assessment of hepatic X-ray attenuation, which is related to liver fat content. However, several factors other than fat (e.g., the presence of iron, copper, glycogen, fibrosis, or edema) confound attenuation values [7]. MRI and MRS have been validated and have been shown to be accurate for detection and quantification of hepatic steatosis. MR-based methods, however, are not used widely for screening because of cost, lack of availability, and lack of validated cutoffs to determine NAFLD. It is therefore appropriate to specify that the technique used to define normal liver fat influences the normal value. As a solution to this problem, in the past few years simple steatosis indexes were developed. These surrogate measurements are based on anthropometric parameters such as body mass index (BMI) and waist circumference (WC) and laboratory tests such as raised liver enzymes (aminotransferases and gamma-glutamyl transpeptidases—ALT, AST, and GGT), altered lipid levels (triglycerides and HDL-cholesterol—TG and C-HDL) and fasting glucose. Overall, steatosis biomarkers can accurately differentiate between the absence and the presence of steatosis [8].

The aim of this study is to evaluate the effect of a lifestyle intervention program on surrogate markers of hepatic steatosis in obese and/or type 2 diabetes patients, enrolled in the C.U.R.I.A.Mo. trial.

## 2. Materials and Methods

### 2.1. Study Design and Subjects

This small, short term non-randomized, non-controlled, retrospective study involves a subsample of subjects who participated in the C.U.R.I.A.Mo. (Centro Universitario di Ricerca Interdipartimentale Attività Motoria) trial. Among 1464 patients, enrolled between January 2010 and February 2014 in the C.U.R.I.A.Mo. trial, we selected 102 cases (56 females and 46 males, aged between 23 and 78, mean = 61, SD = 11.3) according to inclusion criteria that were: granting of written informed consent, aged between 18 and 80 years, BMI ≥ 25 kg/m^2^, diagnosis of obesity or type 2 diabetes with comorbidity. Exclusion criteria were orthopedic or other medical conditions that would contraindicate exercise testing or physical activity. The selection was based on the availability of data needed for the calculation of steatosis indexes. All selected patients participated in the intensive phase of a multidisciplinary lifestyle intervention program at the Healthy Lifestyle Institute of the University of Perugia (C.U.R.I.A.Mo.) [9].

At baseline patients underwent an initial medical visit for screening and collection of clinical data, a physical examination by a specialist in sports medicine, a psychological interview, and an assessment by a dietician. In patients with high-moderate cardiovascular risk, a stress test was performed before starting the intensive lifestyle program. If no exclusion criteria were recorded (orthopedic or other medical conditions that would contraindicate exercise testing or physical activity), patients participated in the three-month intensive lifestyle program. The multidisciplinary clinical trial consisted of an individualized program (groups of five to six patients) of 26 sessions (two/week) of structured indoor exercise, eight sessions of group-based therapeutic education conducted by a doctor of pedagogical sciences and a nutritional program with four sessions of individualized nutritional recording and education, performed by dieticians. The exercise program was performed in a gym and supervised by a graduate in the science and techniques of sports and preventive and adapted physical activity. Each session lasted 90 min: ca. 60 min ca. of aerobic workout and 30 min ca. of circuit training for muscular strength. The aerobic workout was performed using ergometers for cardiovascular work with a gradual increase of the workout intensity (5% every 3 weeks) up to 70% of heart rate reserve. Muscular strength was assessed using isotonic machines starting with a load corresponding to 55% of one repetition maximum; the load was gradually increased every three weeks, if possible.

Laboratory assessment, anthropometric and nutritional data were collected at baseline (T0) and three months after the intensive lifestyle treatment (T1).

Based on the availability of data, six indices related to NAFLD were calculated:The Visceral Adiposity Index (VAI) was calculated using gender-specific equations [10], based on BMI, waist circumference, triglycerides levels expressed in mmol/L and HDL-cholesterol levels expressed in mmol/L.The Fatty Liver Index (FLI), a simple algorithm for the prediction of fatty liver, was calculated based on waist circumference, BMI, γ-GT, triglycerides, expressed in mg/dL [11].The Non-Alcoholic Fatty Liver Disease liver fat score (NAFLD-LFS) and liver fat equation (liver fat), allows identification of NAFLD using easily available clinical and laboratory data (presence of metabolic syndrome; type 2 diabetes; and serum insulin, AST, and ALT) and the individual liver fat content (LIVER FAT), estimated from an equation that contained the same variables as the NAFLD liver fat score [12]. The metabolic syndrome was defined according to criteria of the International Diabetes Federation [13]: central obesity (waist circumference (WC) ≥ 94 cm in men and ≥80 cm in women) and at least 2 of the following factors: (1) serum triglycerides (TG) >150 mg/dL or specific treatment for this lipid abnormality; (2) serum high-density lipoprotein (HDL) cholesterol < 40 mg/dL in men and <50 mg/dL in women or specific treatment for this lipid abnormality; (3) systolic blood pressure (BP) ≥ 130 mmHg or diastolic BP ≥ 85 mmHg or treatment for previously diagnosed hypertension; (4) fasting plasma glucose ≥ 100 mg/dL or previously diagnosed type 2 diabetes.The hepatic steatosis index (HSI) is a simple screening tool for NAFLD calculated according to the following parameters: BMI, ALT/AST ratio, presence or absence of diabetes, and gender, with 2 points added to the algorithm for females [14].The TyG index is measured for the identification of insulin sensitivity, using 2 laboratory parameters: fasting glucose and fasting triglycerides [15].The index may be particularly useful in the presence of steatosis, considering the importance of raised triglycerides in liver fat accumulation. 

Data were analyzed in the whole population sample, in relation to gender, and divided in two subgroups, according to the presence/absence of type 2 diabetes mellitus or “obesity”, that included both obese people as well as overweight patients with metabolic and/or chronic comorbidity (such as dyslipidemia, hypertension, impaired fast glucose). All the data were showed in Table 1.

The Mediterranean Diet Score (MedDietScore) [16] was administered to patients, before and after the lifestyle program, to determine the effect of nutritional intervention on change in adherence to the Mediterranean Diet. In the MedDietScore questionnaire, foods are divided into 11 groups: whole grains, fruits, vegetables, legumes, potatoes, red meat, white meat, milk and dairy products, olive oil, and alcohol. For each item a score ranging from 0 to 5 was assigned, depending on the frequency of consumption of that specific food. Total scores can range from 0 to 55.

The percentage of adherence to the exercise program (calculated as number of sessions executed/ total number of session X 100) was also recorded.

The C.U.R.I.A.Mo. trial has been registered in the Australian New Zealand Clinical Trials Registry, ACTRN12611000255987, and approved by the local Ethics Committee (CEAS Umbria Region, HREC no. 1/10/1633).

### 2.2. Statistical Analysis

Evaluation of normality was conducted with the Kolmogorov–Smirnov test, used to assess of the normal distribution of data. Before analysis, non-normally distributed variables were transformed using Box–Cox transformation, to better approximate the Gaussian distribution. Due to their asymmetry data are shown as median (min/max). 

Changes (Δ scores) were computed for all variables investigated, by subtracting the baseline value (T0) from the 3-months value (T1); *p*-values < 0.05 were regarded as statistically significant. 

Student T-test for paired sample was used to compare all measures (anthropometry, Mediterranean diet adherence score for nutrition habits and adherence to exercise program) before and after treatment (T0 vs. T1), both in the whole sample and in the subgroups according to gender or disease (diabetes and obesity). 

The Pearson correlational coefficients (r) were computed for any relationship between Δ scores of each anthropometric measure, Mediterranean Score measure and adherence to the exercise program.

All analysis were performed with IBM SPSS^®^ version 22.0 (Armonk, NY, USA).

## 3. Results

### Clinical Variables and Steatosis Indexes

Several clinical variables showed statistically significant changes in response to the intensive lifestyle intervention. This was the case for BMI (*p* < 0.0001), which decreased by approximately 0.78 kg/m^2^, for waist circumference (*p* < 0.0001) and triglycerides (*p* = 0.004). The adherence to the Mediterranean diet demonstrated a significant change in nutritional habits (*p* < 0.0001).

The attendance to the exercise program was on average 88.5% (69%/100%).

VAI. After three months of intensive lifestyle treatment, the VAI index improved in the diabetes population (*p* = 0.026).

FLI. The FLI index decreased in the entire population (*p* < 0.0001), as well as in all subgroups with diabetes (*p* < 0.0001) or obesity (*p* < 0.0001), and in both genders (*p* < 0.0001).

Based on the cut-offs reported in the study of Bedogni et al [11] the sample was divided into three groups: 1) NAFLD-free (FLI < 30); 2) possible NAFLD (FLI, 30 to 59), and probable NAFLD (FLI ≥ 60). There were 85 patients with a FLI ≥ 60 at baseline visit, equal to 83% of the total sample, while at T1 this number dropped to 66 (74% of total sample). A separate analysis for patients with estimated steatosis at entry in the trial with FLI > 60 was done. The results indicate, in line with the results of the overall group, an improvement of all indexes after the intervention (TyG, −0.210, *p* < 0.001; Fli, −8.711, *p* < 0.001; Vai, −0.382, *p* = 0.018; HSI, −1.675, *p* = 0.002; NAFLD, −0.853, *p* < 0.001; LiverFat, −1.793, *p* < 0.001; results were showed as estimated difference and p value). In addition, at T1 there was an increase in the number of subjects with FLI < 60: subjects with FLI < 30 increased from 3.9% to 6.9% of the total sample, while subjects with FLI = 30–59 increased from 12.7% up to (28.4%).

NAFLD-LFS. After enrollment in the C.U.R.I.A.Mo. trial, the NAFLD-LFS decreased significantly in the entire population (*p* < 0.0001), as well as in the obesity subgroup (*p* = 0.001), both in males (*p* = 0.006) and in females (*p* = 0.001). The same result was observed for the LIVER FAT equation (*p* = 0.001 for males and *p* < 0.0001 for all other subgroups).

HSI: After three months of intensive lifestyle intervention, HIS significantly decreased in the entire population (*p* < 0.0001), in the obesity subgroup (*p* = 0.001), in males (*p* = 0.036) and in females (*p* = 0.005).

TyG. Data obtained from all patients at the beginning and after the C.U.R.I.A.Mo. trial, have shown a significant change of the TyG index (*p* = 0.002). This improvement was also demonstrated in the diabetes subgroup (*p* = 0.004), and in females (*p* = 0.004).

In the whole sample, we found a significant correlation between changes in waist circumference and ∆HSI (r = 0.278, *p* = 0.005). A significant correlation between change of triglycerides and ∆liver fat score (r = 0.228, *p* = 0.021) was also found. All the results are showed in Table 2 and Table 3.

## 4. Discussion

The lifestyle intervention improved the anthropometric and clinical parameters in the total population, in the obese and/or diabetics. Data collected at baseline and after three months showed significant weight loss, reduced waist circumference, triglycerides reduction and an improvement in eating habits with a change toward higher adherence to the Mediterranean diet. These data demonstrate the clinical efficacy of the C.U.R.I.A.Mo. intervention, a multidisciplinary and combined approach, for the improvement of lifestyle leading to an improvement in health.

Hepatic steatosis indices calculated before and after lifestyle intervention, showed a significant reduction in the total population and in different subgroups (males, females, obesity and diabetes) except for three indices: VAI for the obesity, males and females subgroups, HSI for the diabetes and TyG for obesity.

Recently it was observed that NAFLD patients with metabolic risk factors (among which is the presence of type 2 diabetes) require a larger weight loss (≥10%) to produce the same beneficial effect on histological features of non-alcoholic steatohepatitis, in comparison to subjects with favorable risk factors (eg, absence of diabetes, BMI < 35) [17]. Nonetheless, this amount of weight loss may be achieved, and C.U.R.I.A.Mo. multidisciplinary intervention previously demonstrated its effective impact on body weight [18]. This short-term analysis on steatosis indexes suggests that fat removal occurs rapidly in the course of lifestyle intervention, as also demonstrated by imaging techniques in other settings, both after restrictive diet [19] and after exercise [20]. The correlation analysis suggests some links between steatosis index improvement and waist circumference and triglycerides reductions. This is in line with the scientific literature where a direct association between remission of NAFLD (proven by proton magnetic resonance spectroscopy - and body weight and waist circumference reduction has been consistently demonstrated [21]. All these parameters are in keeping with loss of visceral fat, that has a key role in the pathogenesis of hepatic steatosis.

One of the major limitations of this retrospective study is that the hepatic indexes used only roughly represent the degree of inflammation and lipid infiltration of the liver while the gold standard test for this purpose is histological examination [22,23]. Furthermore, indexes were calculated retrospectively. For this reason, it was not possible to evaluate all the indices in the entire population referred to C.U.R.I.A.Mo., due to the lack of the clinical data necessary for the calculation.

Due to uncertainties surrounding treatment options, screening for hepatic steatosis is not currently performed in the general population and high-risk groups attending diabetes or obesity clinics, such as those who participated in the clinical trial of C.U.R.I.A.Mo., but current Guidelines from European [4] and National association [24] stress the importance of such assessment. Despite the limitations related to their poor specificity, regarding which further studies are needed, the indices of hepatic steatosis are cheap, reproducible, and can be calculated for many people who have access to a basic health plan. Unfortunately, these recommendations were not operative at the time of treatment initiation, and only limited information could be retrospectively derived. In the context of an intensive lifestyle intervention, directed at high metabolic risk populations, hepatic steatosis indexes may be useful in identifying individuals who, after the lifestyle program, still have elevated hepatic indices and must undergo imaging assessment and specific hepato-protective therapy.

To the best of our knowledge this is the only study that has evaluated the performance of six hepatic indices in subjects with high metabolic risk that undergo a multidisciplinary and intensive lifestyle intervention.

Our findings support current recommendations for the management of hepatic steatosis, in which weight loss, obtained through lifestyle modification, represents the first line therapy, even in diabetes subjects. 

In the diabetic population more, and more extensive, investigations on the effect of lifestyle improvement on hepatic steatosis are needed, because people with diabetes are at greater risk of progression of liver disease, developing more severe forms of NAFLD such as steatohepatitis on alcoholic and hepatocellular carcinoma [25,26]. The clinical protocol of C.U.R.I.A.Mo, which consists of the shared and joint action of various professionals, represents a care model aimed at achieving the psychophysical health of the whole person. The growth of awareness in the population at risk and actually maintaining a healthy lifestyle are effective tools in the fight against NAFLD.

## Figures and Tables

**Table 1 jcm-08-00851-t001:** All the features that contribute to the definition of the various indices, at baseline (T0) and after three months (T1) of intensive lifestyle program.

	Overall				Diabetes				Obesity			
	T0	T1	Δ	*p*	T0	T1	Δ	*p*	T0	T1	Δ	*p*
BMI (kg/m^2^)	33.15(22.44/46.03)	32.45(21.9/47.3)	−0.78(−4.8/3.5)	<0.0001	31.8(22.44/46.06)	31.70 (21.9/46.6)	−0,7(−4.8/1.3)	<0.0001	34(26.1/45.2)	33.1(25/47.3)	−0.82(−4.72/3.5)	<0.0001
Waist circumference (cm)	110(85/145)	106.5(84/140)	−3(−39/15)	<0.0001	110(85/133)	106(84/131)	−3(−16/6)	<0.0001	111(91/145)	107(85/140)	−3(−39/15)	<0.0001
Triglycerides (mg/dL)	124(46/591)	116(44/590)	−6(−341/173)	0.004	132(64/591)	116(60/590)	−11(−341/173)	0.007	119(46/452)	116(44/382)	−4(−336/123)	0.152
C-HDL (mg/dL)	50.27(30.16/98.53)	49(32.17/83.50)	−2.01(−25.14/18.10)	0.008	48.77(30.16/76)	48.13(32.17/70.38)	1.01(−21/18.1)	0.626	53.29(31.17/98.53)	49(34.18/83.50)	−4.02(−25.14/11.06)	0.002
GPT	26(7/129)	22(7/68)	−3(118/34)	0.001	22(11/63)	26(7/95)	−2(−64/34)	0.052	27,5(9/129)	22(7/68)	−4(-118/12)	0.006
GOT	22(7/97)	20(7/57)	−2(−77/29)	0.001	22(7/62)	20(9/57)	−3(−42/29)	0.021	22(12/97)	20.5(7/53)	−1(−77/15)	0.025
GGT	25(10/559)	23(8/447)	-3(−112/19)	<0.0001	27(12/372)	24(8/315)	−3(−79/9)	<0.0001	23.5(10/559)	20.5(9/447)	−2.5(−112/19)	0.048

Data (Δ) are expressed as median (min/max). BMI: body mass index; C-HDL: high-density lipoprotein-cholesterol; GPT: serum glutamic pyruvic transaminase; GOT: serum glutamic oxaloacetic transaminase; GGT: gamma-glutamyltransferase.

**Table 2 jcm-08-00851-t002:** Anthropometric, nutritional data and hepatic steatosis indices at baseline (T0) and after three months (T1) of intensive lifestyle program. Data are expressed as median (min/max).

	Overall				Diabetes				Obesity			
	T0	T1	Δ	*p*	T0	T1	Δ	*p*	T0	T1	Δ	*p*
BMI (kg/m^2^)	33.15(22.44/46.03)	32.45(21.9/47.3)	−0.78(−4.8/3.5)	<0.0001	31.8(22.44/46.06)	31.7(21.9/46.6)	−0.7(−4.8/1.3)	<0.0001	34(26.1/45.2)	33.1(25/47.3)	−0.82(−4.72/3.5)	<0.0001
Waist circumference (cm)	110(85/145)	106.5(84/14)	−3(−39/15)	<0.0001	110(85/133)	106(84/131)	−3(−16/6)	<0.0001	111(91/145)	107(85/140)	−3(−39/15)	<0.0001
Triglycerides (mg/dL)	124(46/591)	116(44/590)	−6(−341/173)	0.004	132(64/591)	116(60/590)	−11(−341/173)	0.007	119(46/452)	116(44/382)	−4(−336/123)	0.152
MedDiet (Mediterranean Diet) Score	32(13/38)	37(30/43)	4(−2/24)	<0.0001	27(13/38)	36(30/43)	5(−2/24)	0.006	34(29/36)	37(34/39)	3(−2/9)	0.01
Visceral Adiposity Index	2.01(0.4/10.47)	1.82(0.58/12.9)	−0.05(−8.5/5.63)	0.132	2.1(0.61/8.19)	1.78(0.64/12.9)	−0.2(−3.97/5.63)	0.026	1.84(0.4/10.47)	1.84(0.58/7.85)	0.16(−8.5/2.92)	0.973
Fatty Liver Index	82.78(19.0/99.7)	76.88(19.14/99.42)	−45(−59.19/20.5)	<0.0001	81.9(19.0/99.63)	74.31(19.15/99.16)	−4.15(−39.9/11.03)	<0.0001	86.29(21.31/99.71)	77.05(21.04/99.42)	−5.27(−59.19/20.5)	<0.0001
Non-Alcoholic Fatty Liver Disease liver fat score (NAFLD-LFS)	0.1(−3.16/15.8)	−0.49(−2.76/6.53)	−0.39(−14.94/3.35)	<0.0001	0.8(−2.43/15.8)	0.37(−2.54/6.53)	−0.37(−14.94/3.35)	0.06	−0.82(−3.16/7.60)	−1.26(−2.76/3.31)	−0.4(−8.78/1.25)	0.001
Liver Fat equation	5.42(1.18/26.57)	4.04(1.18/19.72)	−0.8(−22.4/4.98)	<0.0001	7.89(1.41/26.57)	5.94(1.35/19.72)	−1.15(−19.33/4.98)	<0.0001	4.06(1.19/25.97)	3.12(1.18/14.38)	−0.61(−22.4/3.12)	<0.0001
Hepatic Steatosis Index	44.93(31.79/62.77)	43.48(29.81/75.76)	−1.33(−22.14/20.66)	<0.0001	43.68(31.79/62.77)	43.87(29.81/61.67)	−1.13(−22.14/9.52)	0.125	45.68(35.56/59.0)	43.39(34.1/75.76)	−2.06(−17.2/20.66)	0.001
TyG Index	8.87(7.72/11.39)	8.75(7.56/10.72)	−0.93(−1.93/0.81)	0.002	9.04(7.87/11.39)	8.93(7.77/10.72)	−0.2(−1.94/0.81)	0.004	8.61(7.73/10.0)	8.56(7.56/9.81)	−0.03(−1.51/0.8)	0.161

**Table 3 jcm-08-00851-t003:** Anthropometric, nutritional data and hepatic steatosis indices at baseline (T0) and after three months (T1) of intensive lifestyle program. Data are expressed as median (min/max).

	Males				Females			
	T0	T1	Δ	*p*	T0	T1	Δ	*p*
BMI (kg/m^2^)	32.39(21.4/42.22)	31.7(21.9/41.4)	−0.92(−4.8/1.3)	<0.0001	33.70(26.1/46.06)	32.6(25.00/47.3)	−0.6(−4.72/3.5)	<0.0001
Waistcircumference (cm)	113(92/145)	109(89/140)	−4(−16/6)	<0.0001	109.5(85/133)	104.5(84/131)	−3(−39/15)	<0.0001
Triglycerides (mg/dL)	122(46/6591)	120(44/285)	−5(−341/123)	0.292	131.5(54/452)	111(45/590)	−8(−336/173)	0.003
MedDiet Score	32(13/35)	36(30/39)	5(−2/24)	0.01	33(24/38)	37(34/43)	3.5(−2/17)	0.014
VisceralAdiposity Index	1.70(0.40/8.19)	1.77(0.59/5.36)	−0.05(−3.97/2.92)	0.902	2.27(0.63/10.47)	1.82(058/12.90)	−0.55(−8.50/5.63)	0.05
FattyLiver Index	86.43(21.52/99.70)	80.18(24.09/99.13)	−4(−39.90/12.33)	<0.0001	82.35(19.00/99.63)	74.23(19.14/99.42)	−5.75(−59.19/20.50)	<0.0001
NAFLD-FattyLiver Score	0.56(−2.97/15.80)	−0.39(−2.64/5.30)	−0.36(−14.94/1.52)	0.006	−0.02(−3.16/8.50)	−0.7(−2.76/6.53)	0.44(−8.78/3.35)	0.001
LiverFat Equation	6.79(1.19/26.57)	4.31(1.35/19.72)	−0.72(−19.33/3.46)	0.001	5.34(1.26/25.97)	3.64(1.18/14.38)	−0.83(−22.4/4.98)	<0.0001
HepaticSteatosis Index	42.98(31.79/61.07)	42.13(29.81/75.76)	−1.31(−22.14/20.66)	0.036	45.31(35.56/62.77)	44.23(34.20/62.80)	−1.44(−17.2/15.13)	0.005
TyG index	8.89(7.78/11.39)	8.77(7.56/9.95)	−0.08(−1.94/0.8)	0.115	8.86(7.72/10.52)	8.73(7.70/10.72)	−0.09(−1.51/0.81)	0.004

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
