# Peer review of "Effect of Short Term Intensive Lifestyle Intervention on Hepatic Steatosis Indexes in Adults with Obesity and/or Type 2 Diabetes"

_jcm, 2019, doi:10.3390/jcm8060851_

Round 1
Reviewer 1 Report
This is a small retrospective study performed in a subsample of subjects who participated in larger study enrolled between 2010 and 2014.
The aim of the study was to evaluate the effect of lifestyle intervention program on surrogate markers of hepatic steatosis in three groups: obese, diabetics and overweight (BMI>25) with comorbidity.
Six indexes related to NAFLD were calculated before and after a three-month multidisciplinary lifestyle intervention program.
The authors conclude that the short-term intervention improved the anthropometric and clinical parameters in the population studied as well as in the hepatic steatosis indexes in the total population and in different subgroups.
There are several issues that need to be addressed/clarified to ensure that the data fully support the statements and conclusions made. In particular, I suggest that the Authors give some comments on the following points:
1) This is a small, short term non-randomized, non-controlled, retrospective study.
2) The study was performed in a subsample of 1464 patients where selection criteria have been poorly described.
3) The patients included in the subgroup “obesity” seems too much heterogeneous (morbid obese are considered together with overweight subjects).
4) The very long recruitment time period (2010-2014) could have caused bias selection due to seasonal variations, environmental changes etc.
5) Data on Mediterranean Diet Score and of individual item changes are not reported.
6) The structured indoor exercise program should be better described.
7) Table 3 is missing. In fact, the actual table 3 is a copy of table 2.
8) Based on selection criteria, although liver US was not performed, a group of patients could have been free from steatosis (especially in the group of overweight patients with comorbidities). This makes inappropriate the use of steatosis indexes in the overall population.
9) A separate analysis for patients with estimated steatosis at entry in the trial (i.e. FLI >60) is recommended.
10) Correlations between MED diet score and individual items (if any) should be reported.
11) Changes of steatosis indexes should be correlated with insulin-resistance changes.
12) Data on current drug treatment should be reported.
Author Response
Point 1: This is a small, short term non-randomized, non-controlled, retrospective study.
Response 1: We thank the reviewer for his/her observation. We have added his/her point in the text (line 75)
Point 2: The study was performed in a subsample of 1464 patients where selection criteria have been poorly described.
Response 2: We appreciate the reviewer’s comment. We have now expanded the selection criteria in the text (lines 79-82).
Point 3: The patients included in the subgroup “obesity” seems too much heterogeneous (morbid obese are considered together with overweight subjects).
Response 3: Thank you for this useful observation. While the subgroup obesity contained a wide range of patient weights, it can be noted that study results for this subgroup indicate the impact of the intervention on the general population of this type. We certainly will take this observation into account in future.
Point 4: The very long recruitment time period (2010-2014) could have caused bias selection due to seasonal variations, environmental changes etc.
Response 4: Thanks for your note. We would like to point out that over the years indicated, the selection and treatment of the cases studied followed the same time frames (three cycles/year of three months, with suspension in the warmer summer months); in particular, the sites in which the activities took place were heated and cooled to create consistent conditions.
Point 5: Data on Mediterranean Diet Score and of individual item changes are not reported.
Response 5: Thank you for your suggestion. We give data on Mediterranean Diet Score relating to all group and subgroups studied in this work in tables 2 and 3.
Point 6: The structured indoor exercise program should be better described.
Response 6: Thank you for your suggestion. We have now expanded the description of the structured indoor exercise (lines 95-102).
Point 7: Table 3 is missing. In fact, the actual table 3 is a copy of table 2.
Response 7: We apologize for this oversight. We have now added the correct Table 3.
Point 8: Based on selection criteria, although liver US was not performed, a group of patients could have been free from steatosis (especially in the group of overweight patients with comorbidities). This makes inappropriate the use of steatosis indexes in the overall population.
Response 8: Thank you for your comment. We understand the point made, but also note that liver US have drawbacks such as operator dependability. We will consider this point, however, in future studies.
Point 9: A separate analysis for patients with estimated steatosis at entry in the trial (i.e. FLI >60) is recommended.
Response 9:Thank you for your suggestion. We have followed your advice and have done a separate analysis for patients with estimated steatosis at entry in the trial with FLI >60. The results indicate, in line with the results of the overall group, an improvement of all indexes after the intervention (estimated difference and p value) (TyG,-.210, p<0.001; Fli, -8.711, p<0.001; Vai,-.382, p=0.018; HSI ,-1.675 , p=0.002; NAFLD,-.853, p<0.001; LiverFat, -1.793, p<0.001).< span="">
We have added a line in the text reporting the overall results of these patients (line 177-181).
Point 10: Correlations between MED diet score and individual items (if any) should be reported.
Response 10: We thank the reviewer for his/her suggestion, although correlation analysis was not intended in the first place. Following the reviewer’s suggestion, however, we have done a correlation analysis and we found that there wasn’t significative correlation between delta changes values of steatosis indexes age and delta MED diet score, in all group data. At the same time there wasn’t significative correlation between delta MED diet score and delta changes of BMI and Waist Circumference.
Point 11: Changes of steatosis indexes should be correlated with insulin-resistance changes.
Response 11:Thank you for the suggestion. While we did not measure insulin-resistance parameters in this study, we will consider doing so in future studies.
Point 12: Data on current drug treatment should be reported.
Response 12: Thank you for the precious suggestion: unfortunately, we do not have this data. We will keep it in mind for future research
Reviewer 2 Report
The aim of the study by Reginato et al. was to evaluate the effects of a lifestyle intervention program on surrogate markers of hepatic steatosis and their related clinical and metabolic parameters in obese as well as diabetic patients. This is an interesting paper, which, however, deserves a careful revision.
Specific comments
Introduction
P2, line 46 - drawbacks of liver biopsy: the authors should emphasize that it is not feasible in all NAFLD patients, is costly, and suffers from sampling and intra- and inter-observer variability.
P 2, lines 47-49 - drawbacks of imaging techniques: a) the major problem with hepatic ultrasound is not the cost. Indeed, the authors should emphasize that it is highly operator–dependent, and has limited repeatability and reproducibility (Am. J. Roentgenol., 2007, 189, W320-323) owing to subjective qualitative features of liver echogenicity; b) More precise imaging methodology has been developed including computed tomography (CT), magnetic resonance imaging (MRI) and spectroscopy (MRS); however, they are not widely available. CT provides objective assessment of hepatic X-ray attenuation, which is related to liver fat content. However, several factors other than fat (e.g. the presence of iron, copper, glycogen, fibrosis, or edema) confound attenuation values (J. Magn. Reson. Imaging, 2011, 34,729-749); c) MRI and MRS have been validated and have been shown to be accurate for detection and quantification of hepatic steatosis. At this time, however, MR-based methods are not used widely for screening because of cost, lack of availability, and lack of validated cutoffs to determine NAFLD. The technique used to define normal liver fat influences the normal value.
Materials and Methods
Study design:
- age of selected cases ranged between 23 and 78 years. Please report their mean (SD) or median age. How many were classified as young people, middle age, or elderly people? Inclusion of very different age groups may clarify or confound the impact of the intervention.
- What was the duration of the intensive lifestyle intervention for the selected 102 cases?
Results
- data on NAFLD-FLS , liver fat equation, HIS, FLI and TyG for men and women are not presented in either the text or Tables 2 and 3. Thus it is impossible to evaluate the validity of statistical analysis (e.g. the level of significance according to gender in both obese and diabetes subgroups (P4 beginning line 145 and ending on P 5, line 159). Thus, I suggest to present in tables all the results (BMI, WC, TG, HDL, liver enzymes, Med diet, VAI, FLI, NAFLD-FLS, liver fat equation, HIS, and TyG) before and after 3 months of intensive lifestyle intervention, according to gender.
By the way, table 2 and table 3 are identical.
-Data on BMI, Waist circumference and triglycerides as shown in Table 1 are identical to those reported on both tables 2 and 3.
Author Response
Point 1: P2, line 46 - drawbacks of liver biopsy: the authors should emphasize that it is not feasible in all NAFLD patients, is costly, and suffers from sampling and intra- and inter-observer variability.
Response 1: Thank you for your suggestion, we agree with you. We have added the reviewer’s point in the text (lines 48-51).
Point 2: P2, lines 47-49 - drawbacks of imaging techniques: a) the major problem with hepatic ultrasound is not the cost. Indeed, the authors should emphasize that it is highly operator–dependent, and has limited repeatability and reproducibility (Am. J. Roentgenol., 2007, 189, W320-323) owing to subjective qualitative features of liver echogenicity; b) More precise imaging methodology has been developed including computed tomography (CT), magnetic resonance imaging (MRI) and spectroscopy (MRS); however, they are not widely available. CT provides objective assessment of hepatic X-ray attenuation, which is related to liver fat content. However, several factors other than fat (e.g. the presence of iron, copper, glycogen, fibrosis, or edema) confound attenuation values (J. Magn. Reson. Imaging, 2011, 34,729-749); c) MRI and MRS have been validated and have been shown to be accurate for detection and quantification of hepatic steatosis. At this time, however, MR-based methods are not used widely for screening because of cost, lack of availability, and lack of validated cutoffs to determine NAFLD. The technique used to define normal liver fat influences the normal value.
Response 2: We thank the reviewer very much for this very interesting comment. Accordingly, we have modified the text by adding the reviewer’s comment (lines 53-64 and related references 6 and 7).
Point 3: age of selected cases ranged between 23 and 78 years. Please report their mean (SD) or median age. How many were classified as young people, middle age, or elderly people? Inclusion of very different age groups may clarify or confound the impact of the intervention.
Response 3: Thank you for your suggestion. We now report the mean and standard deviation of the subjects (line 79).
We did not classify subjects as young people, middle age, or elderly people in the text. However, in a correlation analysis we did not find age to be associated with delta score of steatosis indexes, indicating that age groups did not affect the impact of the intervention.
Point 4: What was the duration of the intensive lifestyle intervention for the selected 102 cases?
Response 4: The duration of the intensive lifestyle intervention for the selected 102 cases was based on a three-month intensive lifestyle program. We give this information in the text (line 91).
Point 5: data on NAFLD-FLS , liver fat equation, HIS, FLI and TyG for men and women are not presented in either the text or Tables 2 and 3. Thus it is impossible to evaluate the validity of statistical analysis (e.g. the level of significance according to gender in both obese and diabetes subgroups (P4 beginning line 145 and ending on P 5, line 159). Thus, I suggest to present in tables all the results (BMI, WC, TG, HDL, liver enzymes, Med diet, VAI, FLI, NAFLD-FLS, liver fat equation, HIS, and TyG) before and after 3 months of intensive lifestyle intervention, according to gender.
Response 5: We apologize for the mistake; we presented twice the same table. We now present the correct Table 3 reporting data on NAFLD-FLS , liver fat equation, HIS, FLI and TyG for men and women.
Point 6:By the way, table 2 and table 3 are identical.
Response 6: Thank you for your observation. We have modified the tables
Point 7:Data on BMI, Waist circumference and triglycerides as shown in Table 1 are identical to those reported on both tables 2 and 3.
Response 7: Thank you for your observation. We have modified the tables in part.
Finally, the manuscript have been checked by a native English-speaking colleague.
Round 2
Reviewer 1 Report
The Authors have replied to all the reviewer's requests.